# SAIR: Enabling Deep Learning for Protein-Ligand Interactions with a Synthetic Structural Dataset

**Pablo Lemos**[1] *          **Zane Beckwith**[1] *          **Sasaank Bandi**[1]          **Maarten van Damme**[1]

**Jordan Crivelli-Decker**[1]          **Benjamin J. Shields**[1]          **Thomas Merth**[1]          **Punit K. Jha**[1]

**Nicola De Mitri**[1]          **Tiffany J. Callahan**[1]          **AJ Nish**[2]          **Paul Abruzzo**[2]

**Romelia Salomon-Ferrer**[1]                                        **Martin Ganahl**[1]
                    SandboxAQ[1]                        NVIDIA[2]

## ABSTRACT

Accurate prediction of protein-ligand binding affinities remains a cornerstone problem in drug discovery. While binding affinity is inherently dictated by the 3D structure and dynamics of protein-ligand complexes, current deep learning approaches are limited by the lack of high-quality experimental structures with annotated binding affinities. To address this limitation, we introduce the Structurally Augmented IC50 Repository (**SAIR**), the largest publicly available dataset of protein-ligand 3D structures with associated activity data. The dataset comprises $5,244,285$ structures across $1,048,857$ unique protein-ligand systems, curated from the ChEMBL and BindingDB databases, which were then computationally folded using the Boltz-1x model. We provide a comprehensive characterization of the dataset, including distributional statistics of proteins and ligands, and evaluate the structural fidelity of the folded complexes using PoseBusters. Our analysis reveals that approximately $3\%$ of structures exhibit physical anomalies, predominantly related to internal energy violations. As an initial demonstration, we benchmark several binding affinity prediction methods, including empirical scoring functions (Vina, Vinardo), a 3D convolutional neural network (Onionnet-2), and a graph neural network (AEV-PLIG). While machine learning-based models consistently outperform traditional scoring function methods, neither exhibit a high correlation with ground truth affinities, highlighting the need for models specifically fine-tuned to synthetic structure distributions. This work provides a foundation for developing and evaluating next-generation structure and binding-affinity prediction models and offers insights into the structural and physical underpinnings of protein-ligand interactions. The link to the data will be added upon publication, to preserve anonymity of the submission.

## 1 INTRODUCTION

Understanding the interaction between proteins and ligands is a fundamental problem in chemical biology and drug discovery. The binding affinity of a ligand to its target protein, as well as to off-target proteins, is a critical parameter when it comes to designing small-molecule drugs. In principle, the binding affinity can be derived from the structural information of the protein-ligand complex, as the three-dimensional structure of the complex describes most of the interaction between the protein and the ligand. In practice, however, there are limitations to predicting affinity from structural data, both from an experimental and a computational perspective. From an experimental perspective, despite tremendous advances in the field, there are significant challenges generating experimental

---

*{pablo.lemos,zane.beckwith}@sandboxaq.com

structures for some proteins, limiting the accessibility and resolution of some structures. Additionally, in spite of significant advancement in higher throughput methods, the effort needed to produce experimental structural information makes it difficult to efficiently integrate it into the design cycle. From the computational perspective, despite their tremendous utility, traditional methods used to calculate binding affinities like MM/GBSA (Wang et al., 2019) and free energy methods (e.g. Cournia et al., 2017; Crivelli-Decker et al., 2024; York, 2023) rely on the use of force fields which limit their accuracy, while quantum mechanical based methods remain prohibitively expensive. Additionally, traditional methods strongly depend on the quality of the protein-ligand complex, with small inaccuracies producing large errors in the estimation of binding affinities.

To address these limitations, one approach is to learn surrogate functions that approximate binding affinity directly from protein sequences and ligand SMILES representations (e.g., Öztürk et al., 2019; Jiang et al., 2022; Limbu & Dakshanamurthy, 2022). However, binding affinity is fundamentally determined by the three-dimensional (3D) structure of the protein–ligand complex, which is not fully captured by primary sequence information alone. As a result, deep learning methods that operate on 3D structural inputs—whether using convolutional neural networks (e.g., Jiménez et al., 2018; Zheng et al., 2019; Wang et al., 2021) or graph neural networks (e.g., Son & Kim, 2021)—are generally more accurate and robust.

An important issue for scaling deep learning-based affinity prediction methods utilizing 3D structures is the availability of high-quality crystal structure data and accurate binding affinity values. The number of known protein-ligand structures (both from cryo-EM or from X-ray crystallography) that are paired with measured binding affinity values is fairly limited considering the number of possible combinations that may exist in nature (Askr et al., 2023; Libouban et al., 2023; Wang, 2024; Zeng et al., 2024). Moreover, surrogate models trained on these data are heavily impacted by the quality of crystal structures and accuracy of measured binding affinities used. There has been substantial progress in recent years at improving the coverage of these datasets, however existing datasets still lack sufficient coverage in both protein and ligand space.

One possible solution to address the lack of data coverage, is to augment existing datasets with high-confidence computationally folded structures, through a process termed distillation. This approach is frequently used in the field and has been applied to a number of recent works such as in AlphaFold (Jumper et al., 2021; Abramson et al., 2024), Chai-1 (Chai Discovery, 2024), or NeuralPlexer (Qiao et al., 2024). Other groups have attempted to leverage experimentally determined strucutres by leveraging the PDB and linking them with other external resources like BindingDB. For example, the PLINDER dataset (Durairaj et al., 2024), utilized this approach to curate nearly 450k protein-ligand pairs for both apo and holo structures. However, experimentally derived affinity values were only included if available from BindingDB and only represents a smaller fraction of the full dataset. In another example, the CrossDocked dataset Francoeur et al. (2020) used experimentally solved bound structures and docked bound ligands to other similar binding pockets. However, this dataset doesn't include experimentally verified affinity values and only includes poses with binary classification labels for use in downstream machine learning tasks. Thus, there is a clear need

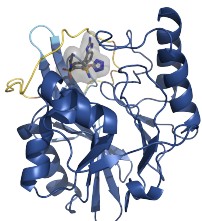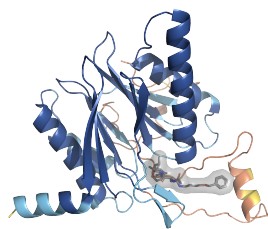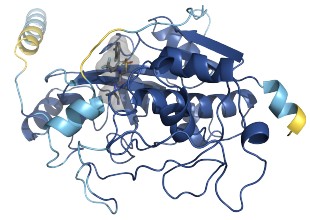

Figure 1: Three example protein-ligand complexes from our **SAIR** dataset. Protein chains are color coded by pLDDT score with blue, yellow, and red regions corresponding to high, medium, and low confidence regions. From left to right the first complex corresponds to sample 4702 with Uniprot ID C7C422 and Ligand InchiKey BFOYHAUOQCJINB-UHFFFAOYSA-N, the second complex corresponds to sample 501640 with Uniprot ID P28062 and Ligand InchiKey ODIWGDQSSAWSIL-GKWCIWIWSA-N, and the third complex corresponds to sample 253438 with Uniprot ID P07858 and Ligand InchiKey SQIHYRIDUCIHLT-HKUYNNGSSA-N.

to further enhance the availability of protein-ligand pairs with bioactivity and binding affinity data to enable the training of large-scale supervised machine learning tasks.

In this work, we present the Structurally Augmented IC50 Repository (**SAIR**): the largest publicly available dataset of protein-ligand 3D structures 1, with annotated potencies (some example structures are shown in Fig. 1). Our data consists of $1,048,857$ protein-ligand complexes, which were obtained from the ChEMBL (Gaulton et al., 2012; Zdrazil et al., 2024) and BindingDB (Liu et al., 2007; 2025) datasets. The data were preprocessed and filtered as described in §2, and the protein–ligand complex structures were folded using the Boltz-1x model (Wohlwend et al., 2024), a publicly available implementation inspired by AlphaFold 3. Results derived from our dataset are presented in §3, including the performance of two binding affinity prediction models: one trained on protein sequence and SMILES representations, and another trained on 3D structural data. We conclude with a summary of key findings in §4.

Our main contribution is the public release of the **SAIR** dataset. The paper describes how the dataset was obtained, and presents some analyses using the data. The link to the data will be added upon acceptance, to preserve the anonymity of the submission.

## 2 DATASET CONSTRUCTION

Table 1: Comparison of Protein-Ligand Binding Databases

| Database | Protein-Ligand Pairs | Structural Data Type | Potency Data |
|---|---|---|---|
| CrossDocked | 22.5m | Synthetic + Experimental | No |
| PDBbind+ (CITE) | 27,385 | Experimental | Yes |
| Binding MOAD | 41,409 | Synthetic Experimental | Yes (15,223 entries) |
| PLINDER | 449,383 | Experimental | Yes (from BindingDB) |
| DockGen | 41,791 | Synthetic | No |
| **SAIR (This Work)** | **1,048,857** | **Synthetic** | **Yes** |

### 2.1 DATASET CURATION

Bioactivity data were obtained from the ChEMBL35 release (Gaulton et al., 2012) and BindingDB (1Q2025) (Liu et al., 2007), and subsequently curated using a minimal set of filters designed to retain a large volume of high-quality data. The specific filters are described below.

**ChEMBL35: 1.** Removed entries missing ligand SMILES or pchembl values. **2.** Removed entries which: did not have a UniProt ID for the protein target, referenced multiple protein targets, or referenced a protein variant. **3.** Removed entries with a data validity comment[1]. **4.** Removed entries where *standard relation* was $<$ or $>$. This step ensures that any measured values obtained are within the limit of detection for the assay. **5.** Only included assays that were flagged by ChEMBL as measuring binding (e.g., $K_i$, IC50, $K_d$). **6.** Removed measurements outside of a reasonable biochemical assay dynamic range (1 pM $< x <$ 100 $\mu$M).

**BindingDB: 1.** Removed entries missing molecule SMILES or IC50 values. **2.** Removed entries without a UniProt ID for the protein target or referenced multiple protein targets. **3.** Removed entries where reported IC50 values contained inequalities (i.e. $<$ or $>$). This step ensures that any measured values obtained are within the limit of detection for the assay. **4.** Remove measurements outside of a reasonable biochemical assay dynamic range (1 pM $< x <$ 100 $\mu$M).

After initial curation, data from both sources were merged into a single table. While this curation strategy can introduce variability in IC50 values by combining data from different assays Landrum & Riniker (2024), this dataset is still fully compatible with the maximal curation strategy outlined in Landrum & Riniker (2024) for data points from ChEMBL. Because BindingDB does not perform curation at the level of specific assays, the maximal curation strategy is not compatible with that source. For protein-ligand complexes that appear in both ChEMBL and BindingDB, we keep the information from both datasets.

---

[1] The data validity comment was introduced in ChEMBL15 and includes information about the quality of the entry and to allow users to make an informed decision on whether to include that value in their analyses (https://chembl.blogspot.com/2020/10/data-checks.html).

All bio-activity values were converted to pIC50 units ($-\log_{10}$). SMILES strings for the ligand molecular structures were standardized by the removal of salts, protonation at neutral pH (where possible), and canonicalization using RDkit. Note that the choice of neutral pH for the ligand protonation is immaterial for the subsequent computational prediction of the protein-ligand structures, as current cofolding models do not predict the positions of hydrogen atoms.

A coarse ligand library filter was applied to exclude likely false positives/false negatives by removing PAINS and molecules with molecular weights exceeding $1250$ Da. Protein-ligand complexes containing proteins with more than 2000 amino acid residues were excluded, in order to increase the probability of successful prediction by the cofolding model on current GPU hardware. Next, duplicate entries were removed based on UniProt accession and canonical SMILES.

The amino acid sequence for each protein was obtained from its UniProt entry using the accession number provided in the ChEMBL or BindingDB dataset. Note that this canonical sequence from UniProt may differ from the one used in the original bioactivity assay. For instance, the experimental protein may have been a truncated construct, a mutant, or a specific quaternary structure (e.g., a homodimer), whereas our analysis used the monomeric sequence from the database.

Finally, to avoid data leakage when using this dataset to train or evaluate models that use structural data from the PDB for training, protein-ligand systems that already have experimentally-solved structures in the PDB were removed. The existence of a corresponding structure in the PDB was determined by finding the Chemical Component Dictionary (CCD) identifier of the ligand (by first computing its InChIKeyHeller et al. (2015) using RDKit) and looking for matches to this (Uniprot ID, CCD ID) pair in the PDB. This search utilized the RCSB GraphQL search APIrcs, and the PDBe REST API provided by EMBL-EBISIF.

This results in $1,048,857$ complexes, with $936,702$ from ChEMBL and $613,597$ from BindingDB (see table in §A). Note the number of complexes from each source adds up to more than the total number, because of complexes that appear in both sources. For structure generation, duplicated complexes were only folded once. The distribution of pIC50 values is shown in Fig. 2a.

## 2.2 STRUCTURE PREDICTION

We used the Boltz-1x folding model to generate 3D structures for all protein-ligand complexes described in §2. Boltz-1 is a publicly available implementation of AlphaFold 3[2]. We selected Boltz-1x over other models, such as AlphaFold 3, primarily to ensure the dataset could be fully open-sourced as a community resource. At the time of generation, the weights and training code for AlphaFold 3 were not available under a license permitting the generation and redistribution of a dataset of this scale ($> 5M$ structures). Boltz-1x is MIT-licensed, ensuring the pipeline remains reproducible. Additionally, Boltz-1x extends the base model by introducing a guiding potential to the diffusion process to prevent clashes, resulting in more physically realistic binding poses. Due to the lack of quaternary structure information, all complexes were treated as monomeric assemblies. We generated five structure samples per complex, as this represents the maximum number we can compute on a single GPU for the longest protein sequence in the dataset (see § below for more details). While it is common practice to increase sample diversity by varying random seeds across multiple runs, we did not apply this technique due to resource constraints[3].

The Boltz-1x model was run using three recycling steps and 200 sampling steps (any other settings are the defaults as of the boltz-1x release). Note that, as all these systems are monomeric, the pairing strategy is irrelevant.

Multiple sequence alignments (MSAs) for input to the model were generated using the MMseqs2 tool (Steinegger, 2017) (via the ColabFold Mirdita et al. (2022) project). This used the UniRef30 sequence database version 2302 and the ColabFoldDB metagenomic sequence database version 202108.

---

[2]There are some minor changes between AlphaFold 3 and Boltz-1, such as the strategy used for Multiple Sequence Alignment (MSA) subsampling.

[3]Further, Boltz-1's MSA subsampling is deterministic with respect to the random seed, unlike other cofolding models such as AlphaFold3 and Chai-1 (Chai Discovery, 2024)), where seed variation is a primary source of stochasticity. As a result, we do not expect significant diversity gains from seed variation in Boltz-1x.

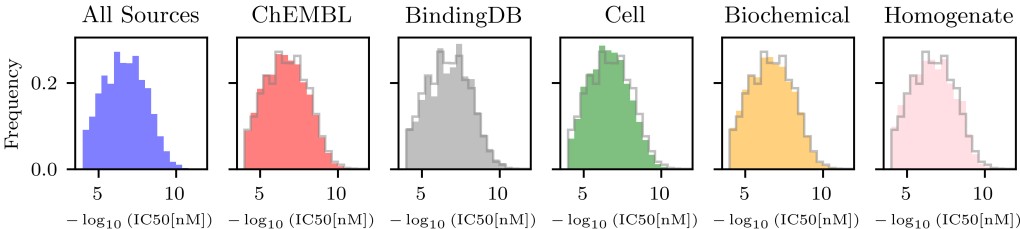

(a) Distribution of pIC50 values stratified by data source and assay type.

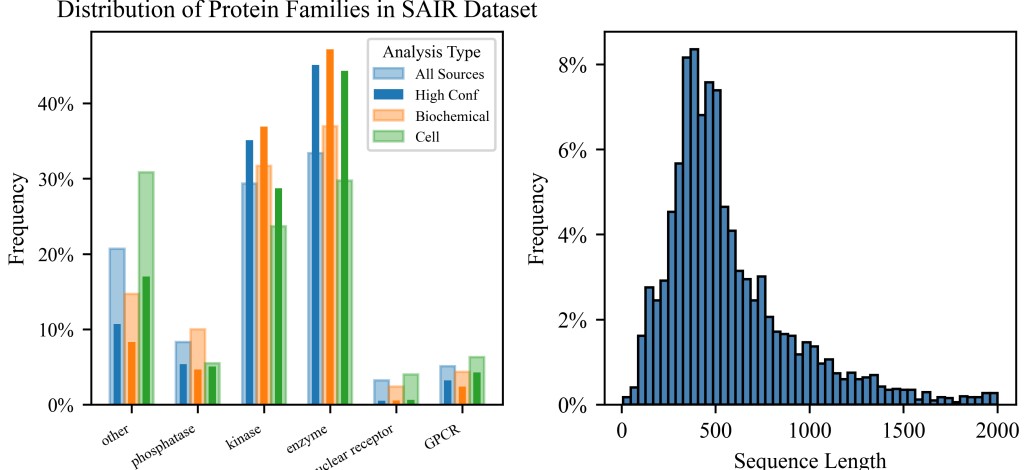

(b) *LEFT*: Distribution of protein families. The thick, shaded bar shows the distribution for the full dataset; whereas the thin solid bar shows the distribution for high confidence predictions only. *RIGHT*: Distribution of protein sequence lengths.

Figure 2: **Dataset Statistics.** (a) pIC50 distribution. For histograms except the first, the overall distribution is shown in grey. (b) Protein family and sequence length distributions.

## 3 RESULTS

### 3.1 DATA STATISTICS

#### 3.1.1 PROTEINS

The $1,048,857$ protein-ligand systems in the dataset comprise $5,149$ unique protein sequences. Of these $5,149$ proteins present in our dataset, $2,150$ are believed to have no structures deposited in the PDB. The distribution of sequence lengths is shown in the right panel of Fig. 2b. Most sequences fall within the 300-500 amino acid range. Beyond 500 residues, the frequency decreases steadily, with very long sequences (e.g., $> 1500$ amino acids) appearing only rarely.

Sequence clustering with MMseqs2 (Steinegger, 2017) using reasonable values for the minimum sequence identity and minimum coverage (MMseqs2 flags `--min-seq-id` and `-c`, respectively) revealed the presence of a large number of singleton clusters. For example, setting `--min-seq-id` and `-c` to $[0.8, 0.8]$, $[0.5, 0.7]$, and $[0.3, 0.2]$ resulted in 3793, 2818, and 1862 clusters, respectively.

Proteins were assigned to a family by using metadata provided by the UniProt database. We first classified them into enzymes or non-enzymes by looking at the presence of an enzymatic activity number (EC number). Enzymes were further subdivided into kinases ($EC = 2.7.x$), phosphatases ($EC = 3.1.x$) and other enzymes. Non-enzymes were subdivided by looking at their gene ontology codes (GO code). For example, the presence of $GO = 0004879$ implies that the protein is a nuclear receptor.

The distribution of protein families accross different assays is shown in the left panel of Fig. 2b. We see that the biochemical assay has a larger proportion of phosphatases, kinases and enzymes, while the cell assay has more nuclear receptors and GPCRs. There is also a larger number of proteins in the

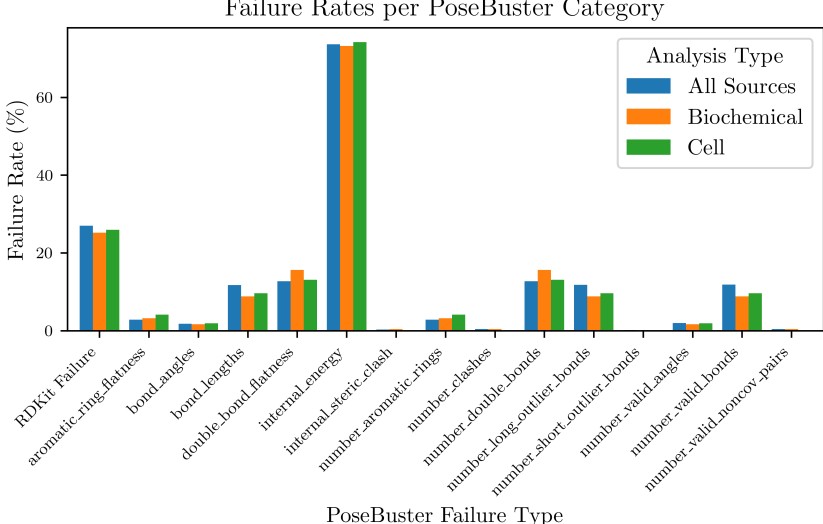

Figure 3: Failure rate for each PoseBusters check, defined as the number of structures failing a given check divided by the total number of failed structures. Bar colors indicate assay type. The first entry in the x-axis corresponds to cases where RDKit failed to load the ligand.

cell assay, for which we could not parse family information. The figure also shows the distribution when we keep only proteins for which the folding model has high confidence, which shows that the model is much more confident in its prediction of kinases and enzymes.

### 3.1.2 LIGANDS

We used `RDKit` (Landrum et al., 2025) to compute a range of chemical descriptors, as summarized in §B. The values capture the statistics of the unique ligands in the dataset.

It is worth noting that we did *not* perform any filtering on the dataset on the basis of things like "drug-likeness" of the ligands, for example filtering samples with ligands below a certain molecular weight. This is to avoid losing useful and chemically-meaningful data points of biologically-relevant species, such as ionic cofactors or small organic fragments that can teach a model protein-small-molecule interaction chemistry. It is our expectation that users will choose to filter samples based on ligand characteristics according to their use case.

### 3.2 POSEBUSTERS

We evaluated all generated protein-ligand structures using PoseBusters (Buttenschoen et al., 2024). A table with summarized results can be found in §C. Overall, Boltz-1x performs well in generating physically valid structures, with only approximately 3% of structures failing any PoseBusters check. This number is consistent with the performance reported in Wohlwend et al. (2024). Notably, only 0.53% of protein-ligand complexes had *all* five generated structures fail. Focusing on different protein families, we find that the model has lower failure rates for kinases, and phosphatases; and fails more often for GPCRs. One interesting case are nuclear receptors, where the overall failure rate is low, but there is a relatively large number of complexes, for which all structures failed (more than for any other family) indicating that some of the nuclear receptor complexes are particularly hard to fold.

For the assemblies that failed PoseBusters validation, we present a more fine-grained analysis of individual test outcomes in Fig. 3. It is important to note that the failure rates shown in Fig. 3 are calculated relative to the $\approx 3\%$ of structures that failed, not the entire dataset. Across all assay types[4] the most frequent source of failure is the internal energy check, which accounts for more than half of all PoseBusters failures. High internal energy often indicates minor steric clashes or bond length anomalies common in raw generative outputs, rather than fundamental structural failures. These

---

[4]We do not show results for the homogenate assay, as there are not enough structures, compared with the rest.

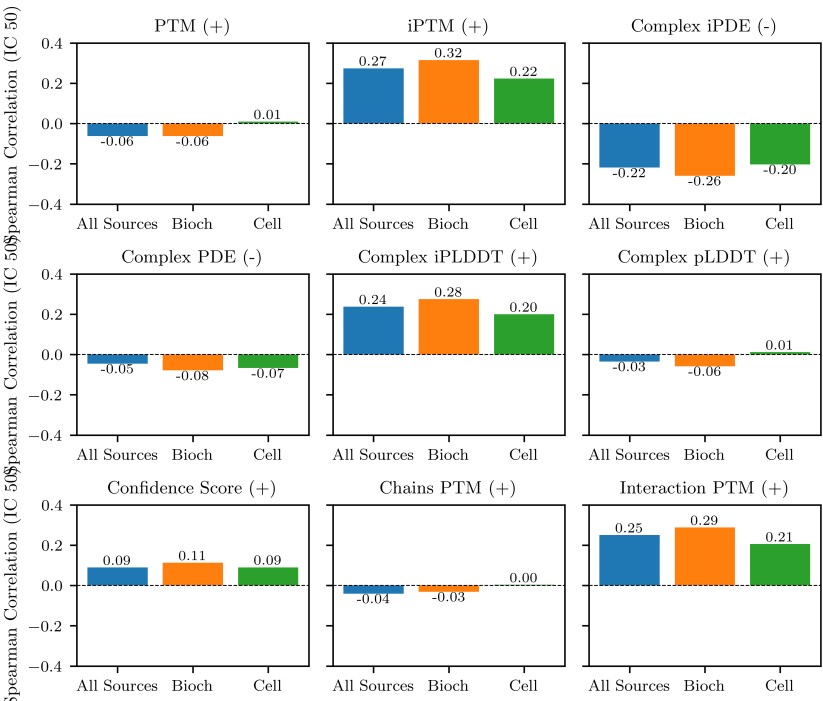

Figure 4: Comparison of the Spearman correlation $r_s$ between different Boltz-1x confidence metrics, and the experimental IC50 activity, by assay type. The signs shown in the title of each panel indicate the expected direction of correlation: negative for PDE and iPDE (as they represent distances), and positive for all other metrics.

can often be resolved via standard energy minimization (e.g., with OpenMM (Eastman et al., 2023)) without altering the binding pose significantly. Other common failure modes include the number of bonds, abnormal bond lengths, and ligands that could not be loaded by RDKit.

## 3.3 BOLTZ CONFIDENCE METRICS

Given that we have access to experimental binding potencies for the corresponding complexes, we assessed whether Boltz-1x's confidence metrics correlate with binding affinity. Prior work has demonstrated that AlphaFold confidence scores correlate with binding affinity in protein–protein interactions (Zambaldi et al., 2024) (PPIs). Here, we explore whether similar correlations exist in the context of protein–ligand interactions. Results are shown in Fig. 4. Focusing on the blue bars (Spearman correlation averaged across all assay types), we observe a significant correlation between certain confidence metrics, particularly those involving the protein-ligand interface —namely iPTM, complex iPDE, and complex iPLDDT—and experimental potency. These findings suggest that Boltz-1x's structural confidence metrics provide some predictive signal for protein–ligand binding affinity. Notably, the strength of the correlation varies by assay type: it is highest for biochemical assays and weakest for cell assays. We hypothesize that this is caused by the great specificity and accuracy of biochemical assays, while cellular and homogenate assays may introduce additional confounding factors such as off-target binding, permeability, and intracellular dynamics. To further probe protein–ligand interaction quality, we introduce a new metric — interaction PTM — defined as the average of the off-diagonal values in the `pair_chains_ptm` confidence head. This metric captures the confidence of the protein with respect to the ligand, and vice versa, and is analogous to the "interaction PAE" described in Zambaldi et al. (2024). We find that interaction PTM exhibits a moderate positive correlation with binding affinity ($r_s = 0.25$), ranking second only to iPTM ($r_s = 0.27$) in predictive power across our dataset.

We can furthermore look at the similarity between generated protein chains and protein chains from the training dataset. We find a high degree of correlation (Spearman correlation of 0.47) between the global PTM confidence and the highest TM-score (normalized by query length) to structures in the training set. This metric is independent of the ligand, which partly explains the poor correlation to the binding affinity. It is better suited for evaluating the global shape of the protein.

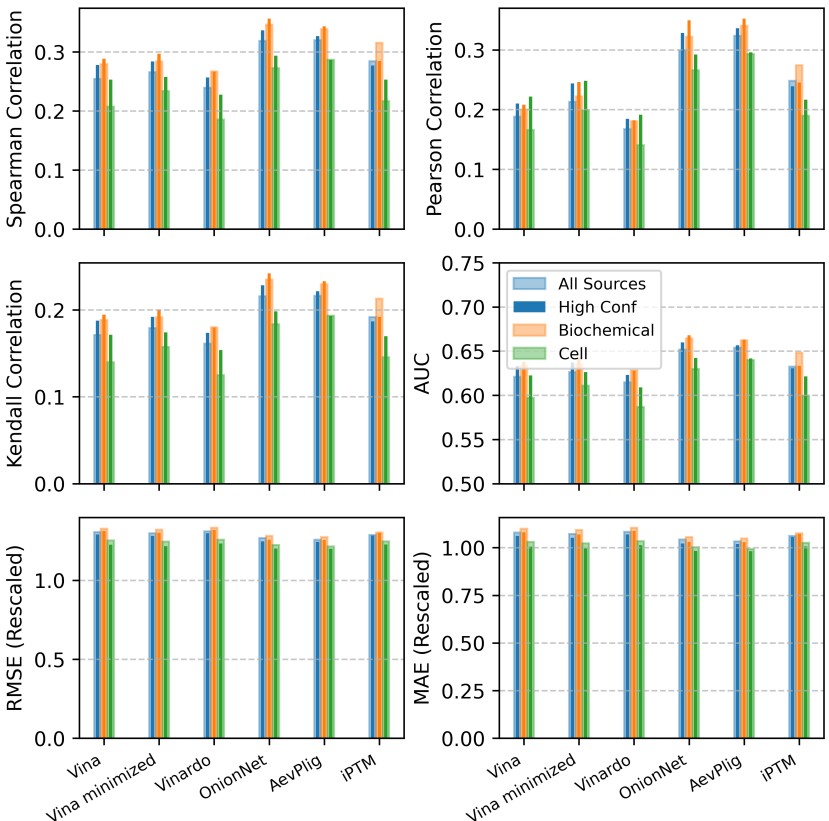

Figure 5: Comparison of binding affinity prediction performance across various methods and assay types. **Top/Middle Rows:** Correlation metrics (Spearman, Pearson, Kendall) and AUC (ranking ability). Higher is better. **Bottom Row:** Error metrics (RMSE, MAE). Lower is better. To ensure a fair comparison between methods outputting pIC50 (ML models) and those outputting energy scores (Vina, Vinardo) or confidence probabilities (iPTM), all predictions were linearly recalibrated to the experimental pIC50 range before calculating RMSE and MAE. **Nested Bars:** The wider, semi-transparent bars represent the full dataset (”All Sources”). The narrower, solid bars nested within them represent the subset of structures where Boltz-1x had high confidence ($> 0.8$).

## 3.4 BINDING AFFINITY MODELS

We use the **SAIR** dataset to benchmark the performance of several binding affinity prediction models. The combination of cofolding-generated structures with structure-based affinity prediction represents a promising and increasingly adopted approach in the scientific community. By providing high-throughput structural models paired with experimentally observed IC50 values, the **SAIR** dataset enables rigorous evaluation of this emerging class of predictive methods.

The field of protein–ligand binding affinity prediction is supported by a vast and diverse body of literature, and a comprehensive comparison of all available methods is beyond the scope of this work. Instead, we focus on three representative and methodologically distinct approaches to binding affinity prediction:

**Empirical scoring functions**: We employ two different traditional empirical scoring functions: AutoDock Vina (henceforth referred to as Vina) (Trott & Olson, 2010) and Vinardo (Quiroga & Villarreal, 2016), both calculated using the `GNINA` library (McNutt et al., 2021). We additionally evaluate first minimizing the ligand pose using the Vina scoring function before scoring the resulting structure, again using Vina (this is referred to as ”Vina minimized”).

**Convolutional neural network** (CNN): As a first method of structure-based machine learning affinity prediction, we employ a three-dimensional CNN, and represent the input by projecting it into 3D voxels. There are various available 3D CNN methods for affinity prediction, but we use Onionnet-2 (Wang et al., 2021), one of the state-of-the-art methods.

**Graph neural network** (GNN): As an alternative structure-based machine learning method, we consider a GNN. GNNs are, theoretically, better suited for the task of affinity prediction, as protein-ligand systems are easily represented as graphs. However, regression from graphs is generally a harder task than regression using voxels, as graph convolutions are non-trivial (Zhang et al., 2019). As our GNN, we use the AEV-PLIG model (Warren et al., 2024; Valsson et al., 2025), which recently showed state-of-the-art performance in structure-based binding affinity prediction.

In all cases (with the exception of the "Vina minimized" approach, as explained above), the given affinity prediction tool is evaluated on the predicted three-dimensional protein-ligand structure as-is.

There are many other methods that could be used for benchmarking binding affinity prediction. For example, the recently developed Boltz-2 (Passaro et al., 2025) accomplishes accurate binding affinity prediction via regression from intermediate embeddings. However, it is trained on similar data to what we present here, therefore it was not used for comparison. On the side of physics-based methods, Free Energy Perturbation (FEP) methods are considered the most accurate, however they are very computationally expensive, making it difficult to use them in a dataset of millions of structures like the one presented in this work.

We use four metrics to compare the performance of the various binding affinity methods. Details about them are shown in §D

We restrict our evaluation to structures derived from ChEMBL, as BindingDB includes experimental protein–ligand complexes that were used in the training of AEV-PLIG. Although the structures in our dataset are synthetically generated and not identical to those used in training, we exclude BindingDB entries to minimize the risk of data leakage and to ensure a fair comparison. As an additional consideration, AEV-PLIG was trained using the BindingNet database Li et al. (2024), which contains synthetically generated structure-activity relationship data derived from ChEMBL. This could account for AEV-PLIG's enhanced performance. Nevertheless, it should be noted that the BindingNet training set represents a minor portion of the overall SAIR database ($\leq 5\%$), and the structures are likely distinct from those in the Boltz computed set.

We present the results of the model comparison in Fig. 5. Across all assay types, the GNN method achieves the highest performance, followed by the CNN method, with the empirical scoring functions performing the worst. However, none of the methods achieve a very high correlation, with Spearman correlations comparable to the ones achieved by some of the interface confidence metrics (even though those were not specifically tuned for binding affinity prediction), such as iPTM, also shown the figure. To compute absolute metrics RMSE and MAE, outputs need to be rescaled, as different methods output values in different units. We therefore apply a linear rescaling to the outputs from each method (see details in §D). We find similar performance accross all methods, with again some slightly better performance by the machine-learning methods. The thinner bars in the figure show the results when we only keep structures for which Boltz-1x predicts a high confidence ($> 0.8$). We find that keeping only these structures improves performance of almost all models, as the structures are more likely to be correct.

It is important to note that both the GNN and CNN models were originally trained on experimental structures, and our evaluation is conducted on synthetic structures generated via cofolding. Fine-tuning these models on a subset of the synthetic dataset would likely improve their performance and better align them with the structural distribution seen at inference time.

As noted in our dataset analysis (Fig. 2b), the dataset contains an uneven distribution of protein families. To ensure that our benchmarking results are not driven solely by well-represented targets (e.g., Kinases), we also performed a family-balanced evaluation. In this analysis, metrics are calculated using sample weights inversely proportional to the frequency of the protein family. As detailed in §G, we observe that the performance ranking of models and the magnitude of correlations remain consistent under this balanced evaluation (Fig. 8), providing empirical evidence that the predictive signal generalizes across diverse protein targets.

## 3.5 TRAINING MODELS ON **SAIR**

A primary motivation for creating **SAIR** is to address the scarcity of structure-affinity data available for training deep learning models. While the benchmarks in Fig. 5 assess the zero-shot performance of existing models on our dataset, a critical question is whether **SAIR** can serve as an effective training resource to improve model generalization.

Recent independent work by Wei et al. (2025) provides strong empirical validation of **SAIR**'s utility in this context. In their development of `GatorAffinity`, a geometric deep learning scoring function, the authors utilized over one million protein-ligand complexes from **SAIR** for large-scale pre-training before fine-tuning on experimental structures. Their analysis revealed two key findings relevant to our dataset:

**Data Scaling Law:** The study demonstrated a power-law relationship between the size of the synthetic pre-training data and downstream model performance. Specifically, augmenting their training set with **SAIR** consistently reduced the Root Mean Square Error (RMSE) on the independent PDB-bind benchmark, with the best performance achieved when combining **SAIR**'s $IC_{50}$ data with $K_d$ and $K_i$ datasets.

**State-of-the-Art Performance:** By leveraging the scale of **SAIR**, the GatorAffinity model was able to outperform existing state-of-the-art methods, including GIGN and PSICHIC, achieving an RMSE of 1.293 on the PDBbind benchmark compared to 1.343 without the use of **SAIR**.

These results confirm that despite the synthetic nature of the structures, **SAIR** contains a strong, learnable signal that materially improves the performance of structure-based affinity prediction models when used for pre-training.

## 4 CONCLUSIONS

In this work, we introduce the Structurally Augmented IC50 Repository (**SAIR**), a large-scale dataset of protein–ligand 3D structures paired with annotated binding affinities. Comprising $5,244,285$ synthetically-generated structures representing $1,048,857$ protein-ligand complexes, each annotated with experimentally-determined potency, the dataset is designed to significantly expand the volume of data available for training and evaluating structure-based deep learning models in drug discovery.

We rigorously evaluated the quality of the generated structures using PoseBusters and observed a low overall failure rate of approximately $3\%$. To assess the utility of the dataset for predictive modeling, we benchmarked several structure-based binding affinity prediction methods. Graph neural networks performed best, followed by convolutional neural networks and empirical scoring function methods. However, all models achieved only modest correlations, comparable to those obtained from the folding model's interface confidence metrics. This suggests that models trained on experimental structures may not generalize well to synthetic data, highlighting the potential need for fine-tuning on generated complexes.

Looking ahead, this work opens several promising avenues for future research. We also aim to use **SAIR** to improve the performance of binding affinity prediction methods. More broadly, fine-tuning existing affinity prediction models—or developing new architectures specifically optimized for synthetic protein–ligand complexes—could lead to significant gains in predictive accuracy. Beyond affinity prediction, the dataset may also support self-distillation strategies for cofolding models, as has been demonstrated in other data modalities.

Finally, **SAIR** represents a valuable resource for inverse design tasks, enabling generative approaches to create new ligands conditioned on a target protein—further expanding the possibilities for structure-based drug discovery.

**Limitations**: While **SAIR** provides an unprecedented scale of data, it has inherent limitations. First, as a distilled dataset, it carries the inductive biases of the teacher model, Boltz-1x. To assist users in identifying potential mode collapse or model bias, we have included "pocket diversity" statistics in the dataset metadata (see §F), allowing users to filter out proteins where the model fails to generate diverse binding modes.

Second, validation via PoseBusters ensures chemical plausibility (e.g., valid bond lengths, low energy), but it does not guarantee biological correctness. Because we explicitly filtered out proteins present in the PDB to prevent data leakage, ground truth structures are not available for RMSD comparisons. Users should treat the provided confidence metrics (iPTM, iPLDDT) as proxies for biological plausibility rather than guarantees.

Finally, in this version of the dataset, we folded the monomeric protein chain and the ligand of interest; cofactors and ions were not explicitly modeled. To mitigate this, we have added a `drug_like` filter column to the dataset metadata, allowing users to exclude tiny ligands, ions, or fragments that may be artifacts or cofactor-dependent.

## ETHICS STATEMENT

The dataset is curated from publicly available, open-access databases, specifically ChEMBL and BindingDB, which contain experimentally determined bioactivity data. We have not used any private or sensitive human data. The primary purpose of this dataset is to advance drug discovery through computational methods, a field that aims to develop new treatments for diseases. The potential societal benefits are significant and do not involve foreseeable harm.

The dataset, a link to which will be provided upon acceptance, will be released publicly to promote open science and collaboration in the research community. We believe that providing this resource to the public will foster a collaborative environment and accelerate scientific progress.

## REPRODUCIBILITY STATEMENT

To ensure the reproducibility of our work, we have provided comprehensive details on the dataset's construction, model evaluation, and analyses in the main paper and its appendices.

Furthermore, for full transparency and to aid in future research, we have provided a detailed manifest of the dataset's file organization and a description of each column in the central `sair.parquet` file in §F. Upon acceptance, the link to the dataset will be made publicly available, allowing other researchers to replicate our findings and build upon our work

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

Table 2: Data distribution by source and assay. Note that the total number between both assays is larger than the number of generated structures. That is because of protein-ligand pairs that appear in both datasets

| Assay | Source | |
|---|---|---|
| | **BindingDB** | **ChEMBL** |
| biochem | 2,293 | 416,331 |
| cell | 0 | 183,286 |
| homogenate | 0 | 13,980 |
| na | 934,409 | 0 |
| **Total** | **936,702** | **613,597** |

Barbara Zdrazil, Eloy Felix, Fiona Hunter, Emma J Manners, James Blackshaw, Sybilla Corbett, Marleen de Veij, Harris Ioannidis, David Mendez Lopez, Juan F Mosquera, et al. The chembl database in 2023: a drug discovery platform spanning multiple bioactivity data types and time periods. *Nucleic acids research*, 52(D1):D1180–D1192, 2024.

Xin Zeng, Shu-Juan Li, Shuang-Qing Lv, Meng-Liang Wen, and Yi Li. A comprehensive review of the recent advances on predicting drug-target affinity based on deep learning. *Frontiers in Pharmacology*, 15:1375522, 2024.

Si Zhang, Hanghang Tong, Jiejun Xu, and Ross Maciejewski. Graph convolutional networks: a comprehensive review. *Computational Social Networks*, 6(1):1–23, 2019.

Liangzhen Zheng, Jingrong Fan, and Yuguang Mu. Onionnet: a multiple-layer intermolecular-contact-based convolutional neural network for protein–ligand binding affinity prediction. *ACS omega*, 4(14):15956–15965, 2019.

HARDWARE SPECIFICATIONS

This work was performed on an H100 cluster in DGX Cloud on Google Cloud provisioned by NVIDIA's AI Accelerator team. This was initially provisioned as a managed 35-Node (280 GPUs) cluster, and was non-disruptively scaled up to 96 nodes (760 GPUs) to allow for scale-out workloads.

In total, these calculations used approximately 130k GPU-hours of compute time on this cluster.

## A  DATA DISTRIBUTION

§A shows the distribution of datapoints by source and assay.

## B  CHEMICAL DESCRIPTORS

Table 3 shows a summary of the chemical descriptors of the different ligands in our dataset.

## C  POSEBUSTERS RESULTS

Table 4 summarizes the posebusters results accross different protein families.

## D  METRICS

In §3.4, we use the following metrics to compare the various affinity prediction models.

- **Spearman Correlation**: The Spearman correlation between the predicted and experimental binding affinity, defined as:

$$r_s = 1 - \frac{6 \sum d_i^2}{n(n^2 - 1)} \tag{1}$$

  where $d_i$ is the difference between the ranks of the predicted and experimental binding affinities, and $n$ is the number of samples.

Table 3: Chemical descriptors across the dataset (aggregation of unique ligand entries).

| Descriptor | Min | Max | Mean | Stddev. |
|---|---|---|---|---|
| Molecular weight | 17.0 | 1.25e+03 | 4.46e+02 | 1.25e+02 |
| Heavy atom count | 1.00 | 94.0 | 31.7 | 8.88 |
| Hetero atom count | 0.00 | 46.0 | 8.55 | 3.31 |
| H-bond acceptor count | 0.00 | 36.0 | 5.89 | 2.33 |
| H-bond donor count | 0.00 | 25.0 | 1.89 | 1.58 |
| Topological polar surface area | 0.00 | 6.40e+02 | 92.9 | 43.3 |
| Wildman-Crippen LogP | -13.1 | 19.2 | 3.80 | 1.72 |
| QED (Drug-likeness) | 0.00684 | 0.948 | 0.498 | 0.204 |
| Rotatable bond count | 0.00 | 53.0 | 6.11 | 3.69 |
| Fraction Csp3 | 0.00 | 1.00 | 0.321 | 0.177 |
| Aliphatic carbocycle count | 0.00 | 19.0 | 0.328 | 0.720 |
| Aliphatic heterocycle count | 0.00 | 20.0 | 0.754 | 0.828 |
| Aliphatic rings count | 0.00 | 21.0 | 1.08 | 1.08 |
| Aromatic carbocycle count | 0.00 | 20.0 | 1.55 | 0.935 |
| Aromatic heterocycles count | 0.00 | 11.0 | 1.45 | 1.10 |
| Aromatic ring count | 0.00 | 20.0 | 3.00 | 1.16 |
| Bridgehead atom count | 0.00 | 20.0 | 0.110 | 0.598 |
| Spiro atom count | 0.00 | 6.00 | 0.0392 | 0.210 |

Table 4: Summary of PoseBusters results by family

| Family | Metric | Structures | Assemblies |
|---|---|---|---|
| **Overall Analysis** | Total failed | 166,241 | 5,526 |
| | Total | 5,244,285 | 1,048,857 |
| | Percentage failed | 3.17% | 0.53% |
| **Enzyme** | Total failed | 59,169 | 1,944 |
| | Total | 1,699,025 | 339,805 |
| | Percentage failed | 3.48% | 0.57% |
| **Kinase** | Total failed | 34,160 | 1,639 |
| | Total | 1,531,115 | 306,223 |
| | Percentage failed | 2.23% | 0.54% |
| **Other** | Total failed | 46,248 | 1,200 |
| | Total | 1,170,445 | 234,089 |
| | Percentage failed | 3.95% | 0.51% |
| **Phosphatase** | Total failed | 12,181 | 329 |
| | Total | 418,895 | 83,779 |
| | Percentage failed | 2.91% | 0.39% |
| **GPCR** | Total failed | 10,304 | 188 |
| | Total | 267,650 | 53,530 |
| | Percentage failed | 3.85% | 0.35% |
| **Nuclear Receptor** | Total failed | 4,179 | 226 |
| | Total | 157,155 | 31,431 |
| | Percentage failed | 2.66% | 0.72% |

- **Pearson Correlation**: The Pearson correlation between the predicted and experimental binding affinity, defined as:

$$r_p = \frac{\sum(x_i - \bar{x})(y_i - \bar{y})}{\sqrt{\sum(x_i - \bar{x})^2}\sqrt{\sum(y_i - \bar{y})^2}} \tag{2}$$

where $x_i$ and $y_i$ are the predicted and experimental binding affinities, respectively, and $\bar{x}$ and $\bar{y}$ are the means of the predicted and experimental binding affinities, respectively.

- **Kendall's Tau**: Kendall's Tau is a measure of the ordinal association between two quantities, defined as:

$$\tau = \frac{(n_c - n_d)}{\frac{1}{2}n(n-1)} \tag{3}$$

  where $n_c$ is the number of concordant pairs, and $n_d$ is the number of discordant pairs, and $n$ is the number of samples.

- **Area Under the Curve (AUC)**: The AUC is a measure of the ability of a model to distinguish between positive and negative samples. It is defined as the area under the Receiver Operating Characteristic (ROC) curve, which is a plot of the true positive rate against the false positive rate. To calculate the AUC, we first need to define a threshold for the predicted binding affinity, and then compute the true positive rate (TPR) and false positive rate (FPR) for that threshold. We use a threshold of $100nM$, which is a common threshold for binding affinity prediction.

- **Root Mean Squared Error (RMSE)**: Measures the square root of the average squared differences between predicted and actual values:

$$\text{RMSE} = \sqrt{\frac{1}{n}\sum_{i=1}^{n}(y_i - \hat{y}_i)^2} \tag{4}$$

- **Mean Absolute Error (MAE)**: Measures the average magnitude of errors:

$$\text{MAE} = \frac{1}{n}\sum_{i=1}^{n}|y_i - \hat{y}_i| \tag{5}$$

A direct comparison of error metrics (RMSE and MAE) is complicated by the differing output units of the benchmarked methods. Deep learning models (AEV-PLIG, OnionNet-2) predict pIC50 values directly, whereas empirical scoring functions output binding free energies (Vina, Vinardo in $\text{kcal/mol}$) or confidence probabilities (iPTM $\in [0, 1]$). To enable a fair comparison of predictive power, we applied a linear recalibration to all methods before calculating RMSE and MAE. For each method, we fit a simple linear regression model $\hat{y} = \alpha x + \beta$ to map the raw model scores $x$ to the experimental pIC50 values $y$. This removes systematic shifts and scaling differences (e.g., mapping kcal/mol to pIC50), ensuring that the reported error metrics reflect the model's ability to capture the underlying affinity signal rather than its unit scaling.

## E  POCKET DIVERSITY

We can use our model to gain insight into the effect that changing the input ligand has in the generated protein conformation. If we give Boltz-1x sufficiently distinct ligands, is the model able to detect different appropriate binding sites, or will it re-use the pockets it has seen during training?

To address this, we need to define a pocket. First, we define pocket residue as every residue that has a non-hydrogen atom within $6\mathring{A}$ to the closest ligand atom, a cutoff commonly adopted in the field. The set of pocket residues defines a pocket, and two pockets are considered similar if

$$\frac{|p_1 \cap p_2|}{\min(|p_1|, |p_2|)} \geq \text{threshold}, \tag{6}$$

where we defined the threshold to be $0.8$. That allows us to cluster the detected pockets per protein into groups, such that no group shares a similar pocket.

Fig. 6 shows both the diversity in pockets over the five generated structures per protein/ligand complex (left panel), and the diversity for a given protein as we change ligands (right). AlphaFold3-like models are known to generate similar conformations for different diffusion samples, which we also find when looking at the pocket diversity, with most complexes generating ligands in the same pocket for all five generated samples.

However when looking at a protein chain, with different ligands, we find a significant fraction of systems with a variable set of pockets. The most extreme example of this is protein P10636, where we found well over a thousand different potential binding sites for 345 different ligands.

This shows a potential reason for the fat tail in the number of distinct pockets per protein. When Boltz-1x is uncertain about the structure or if the protein is very flexible, then we find a very diverse set of protein conformations. The pocket-residues will similarly change a lot, and there is no well defined binding site.

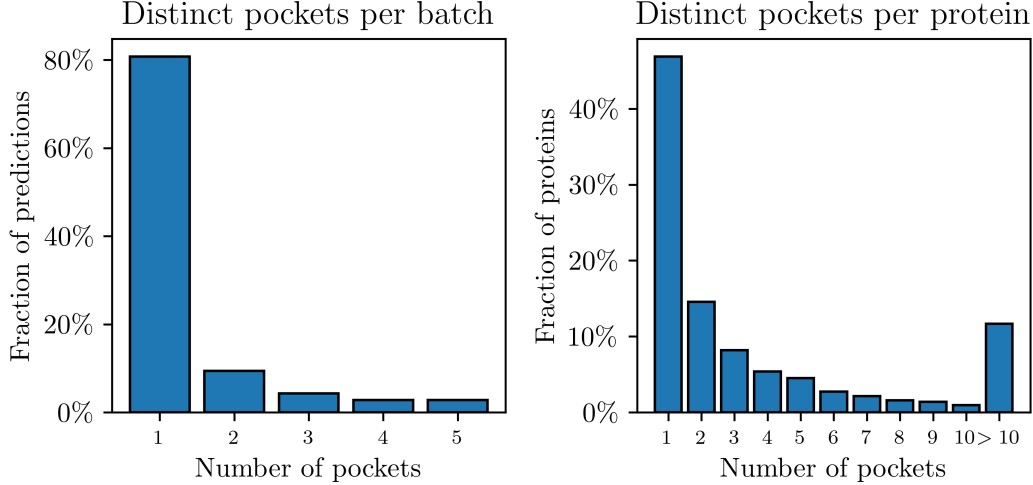

Figure 6: Pocket diversity for different samples and different ligands. Pocket similarity was determined using Eq. (6). **Left**: The diversity of pockets in the five generated structures per protein/ligand complex. **Right**: The diversity of pockets for different proteins.

We can also study pocket similarity to the training set. Fig. 7 shows the result of performing a similarity search of all generated structures against the Boltz-1x training data, the pocket-LDDT score, as defined in Durairaj et al. (2024). The pocket-LDDT is defined by structurally aligning predicted structures to ground truth structures, and calculating the average LDDT over the backbone carbon atoms in the aligned pocket residues. In our case, this score is not a measure of correctness, but more a measure of similarity to the training dataset.

We find no correlation between the pocket-LDDT and the interface confidence (Spearman correlation of -0.02), but we do find most pockets to be highly similar to the training data. We have previously seen that Boltz is able to generate multiple distinct binding poses, which together implies that Boltz successfully places ligand atoms in plausible looking pockets.

## F  IC50 DATASET STRUCTURE AND CONTENTS

This appendix provides a detailed description of the **SAIR** dataset, its file organization, and the contents of its primary data files.

### F.1  DATASET MANIFEST

The IC50 dataset is organized into the following primary components:

- **`sair.parquet`**: This is the central dataframe of the dataset, containing all curated IC50 data, associated original source metadata, results from PoseBusters structural validity checks, Boltz-1x prediction confidence measures, and other relevant metadata for each protein-ligand complex.
- **`structures/`**: This directory contains all the predicted 3D structures generated by the Boltz-1x co-folding model. For each unique protein-ligand complex, five distinct predicted structures (referred to as "models") are provided. Each structure is stored as a `.cif` file, named according to the convention `sample_<entry id>_model_<model>.cif`. Here, `entry id` corresponds to the `entry_id` field in `sair.parquet`, and `model` identifies the specific sample (0 through 4) from the co-folding model.

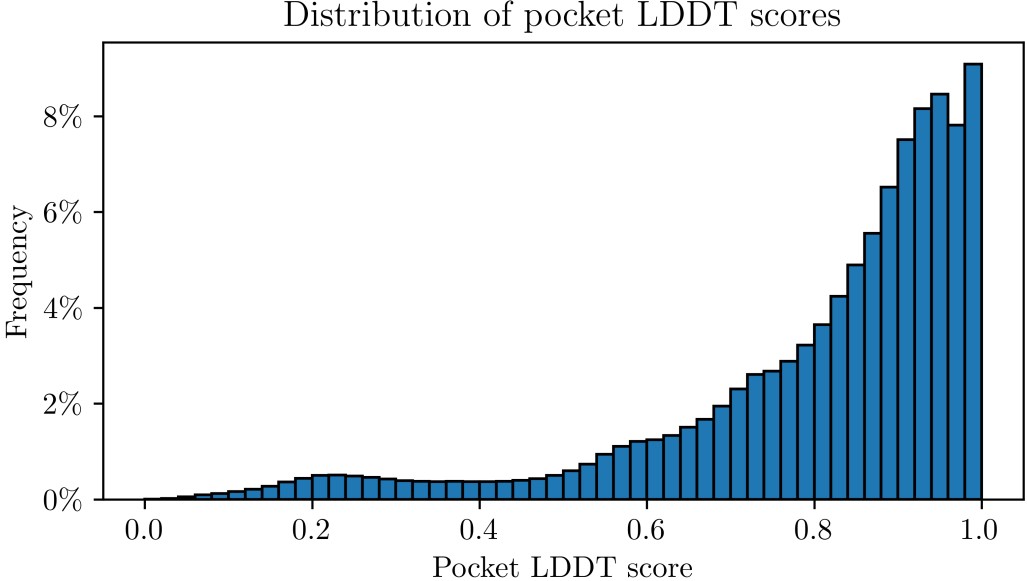

Figure 7: The distribution of pocket LDDT's calculated, comparing the generated structures to the training dataset.

- **prediction_confidences/**: This directory stores the raw `.json` confidence files directly produced by the Boltz-1x model. The files follow the naming convention `confidence_sample_<entry id>_model_<model>.json`, with `entry id` and `model` consistent with the structure files. These files contain detailed confidence metrics beyond those summarized in `sair.parquet`.

- **prediction_plddts/**: This directory contains the raw `.npz` files for Predicted Local Distance Difference Test (pLDDT) values, also output directly by the Boltz-1x model. These files are named `plddt_<entry id>_model_<model>.npz`, using the same `entry id` and `model` identifiers. pLDDT values provide residue-level confidence estimates for the predicted structures.

## F.2 SAIR.PARQUET COLUMN DESCRIPTIONS

The `sair.parquet` file serves as the main tabular data source, consolidating key information for each structure-potency pair.Table 5, Table 6, Table 7 and Table 8; describe each column in `sair.parquet`, categorized for clarity.

## G ROBUSTNESS ANALYSIS

To address potential concerns regarding distillation bias—specifically, that the model might perform well only on protein families that are over-represented in the dataset or high-confidence subset (such as Kinases)—we conducted a robustness check using family-balanced metrics.

Fig. 8 presents the performance of all benchmarked models when evaluated with sample weights inversely proportional to the frequency of the corresponding protein family. Comparing these results to the standard benchmarks in Fig. 5, we observe consistent trends: machine learning models (AEV-PLIG, OnionNet-2) continue to outperform empirical scoring functions, and confidence metrics (iPTM) maintain their correlation with affinity. This stability indicates that the reported performance is not an artifact of dataset imbalance.

Table 5: Description of Columns in `sair.parquet`: Identifiers and Inputs

| Column Name | Description |
|---|---|
| **Identifiers** | |
| entry_id | Unique identifier for the protein-ligand complex, common across all 5 predicted models for that complex. |
| index | Unique identifier for the sample for this entry_id, ranging from 0 to 4 (included). |
| **Inputs** | |
| protein | UniProt accession ID of the protein target. |
| sequence | Amino acid sequence of the protein. |
| SMILES | Sanitized SMILES string of the ligand, used for structure generation. |
| srcSMILES | Original SMILES string of the ligand as provided in the raw underlying data source (ChEMBL or BindingDB). |

Table 6: Description of Columns in `sair.parquet`: Potency Data and Metadata

| Column Name | Description |
|---|---|
| **Potency Data and Metadata** | |
| source | Origin of the bioactivity data: 'ChEMBL' or 'BindingDB'. |
| description | Description of the assay as provided by the source database. |
| potency | Original potency value (e.g., IC50, Ki, Kd) from the underlying data source in its native units. |
| assay_type | Type of assay (e.g., 'biochemical', 'cell', 'homogenate', or 'na'). |
| assay_id | The original ChEMBL assay ID (if available) for granular grouping. |
| pIC50 | Potency value converted to negative-log10 units. |
| family | The protein family classification for the target protein. |
| drug_like | Boolean flag indicating if the ligand is drug-like (excludes ions/cofactors). |
| pocket_diversity | Metric indicating the diversity of binding modes generated for this protein. |

Table 7: Description of Columns in `sair.parquet`: Binding Affinity Predictions and Model Confidence

| Column Name | Description |
|---|---|
| **Binding Affinity Prediction Results** | |
| vinardo_score | Vinardo docking score for the predicted structure. |
| vina_score | Vina docking score for the predicted structure. |
| vina_score_min | Vina docking score for the predicted structure, after minimization. |
| aevplig_score | Predicted binding affinity score from the AEV-PLIG model. |
| onionnet_score | Predicted binding affinity score from the Onionnet-2 model. |
| **Model Confidence Estimates** | |
| confidence_score | Overall confidence score produced by the Boltz-1x model for the predicted structure. |
| chains_ptm | Predicted TM-score (PTM) specifically for the interaction between different chains. |
| ptm | Predicted TM-score, a global metric of structural accuracy. |
| iptm | Interface Predicted TM-score, focusing on the accuracy of the protein-ligand interface. |
| complex_pde | Complex Predicted Distance Error. |
| complex_ipde | Complex Interface Predicted Distance Error, focusing on errors at the interface. |
| complex_plddt | Predicted Local Distance Difference Test (pLDDT) for the complex. |
| complex_iplddt | Interface Predicted Local Distance Difference Test (iPLDDT) for the complex. |
| interaction_ptm | Custom metric: Average of off-diagonal terms in the pair chains PTM confidence head (protein-ligand interaction confidence). |

Table 8: Description of Columns in `sair.parquet`: PoseBusters Results

| Column Name | Description |
| --- | --- |
| **PoseBusters Results** | (Boolean flags indicate 'true' for pass/success, 'false' for fail/issue, counts are integers) |
| `mol_pred_loaded` | Indicates if the predicted ligand molecule could be successfully loaded by RDKit. |
| `sanitization` | Indicates if the ligand molecule passed RDKit's sanitization checks. |
| `inchi_convertible` | Indicates if the ligand molecule is convertible to an InChI string. |
| `all_atoms_connected` | Indicates if all atoms in the ligand are connected (no disconnected fragments). |
| `bond_lengths` | Indicates if all ligand bond lengths are within expected ranges. |
| `bond_angles` | Indicates if all ligand bond angles are within expected ranges. |
| `internal_steric_clash` | Indicates the presence of internal steric clashes within the ligand. |
| `aromatic_ring_flatness` | Indicates if aromatic rings in the ligand maintain expected flatness. |
| `double_bond_flatness` | Indicates if double bonds in the ligand maintain expected flatness. |
| `internal_energy` | Indicates if the ligand's internal energy is within reasonable bounds (passed internal energy checks). |
| `mol_cond_loaded` | Indicates if the conditioned (input) molecule could be loaded. |
| `passes_valence_checks` | Indicates if the ligand passes valence rules. |
| `passes_kekulization` | Indicates if the ligand can be successfully kekulized by RDKit. |
| `number_clashes` | The count of clashes detected between the protein and ligand. |
| `number_short_outlier_bonds` | The count of bonds in the ligand that are unusually short. |
| `number_long_outlier_bonds` | The count of bonds in the ligand that are unusually long. |
| `number_valid_bonds` | The count of valid bonds in the ligand. |
| `number_valid_angles` | The count of valid bond angles in the ligand. |
| `number_valid_noncov_pairs` | The count of valid non-covalent pairs between protein and ligand. |
| `number_aromatic_rings` | The count of aromatic rings in the ligand. |
| `number_double_bonds` | The count of double bonds in the ligand. |
| `all_passed` | Boolean flag indicating whether the structure passed all PoseBusters checks. |

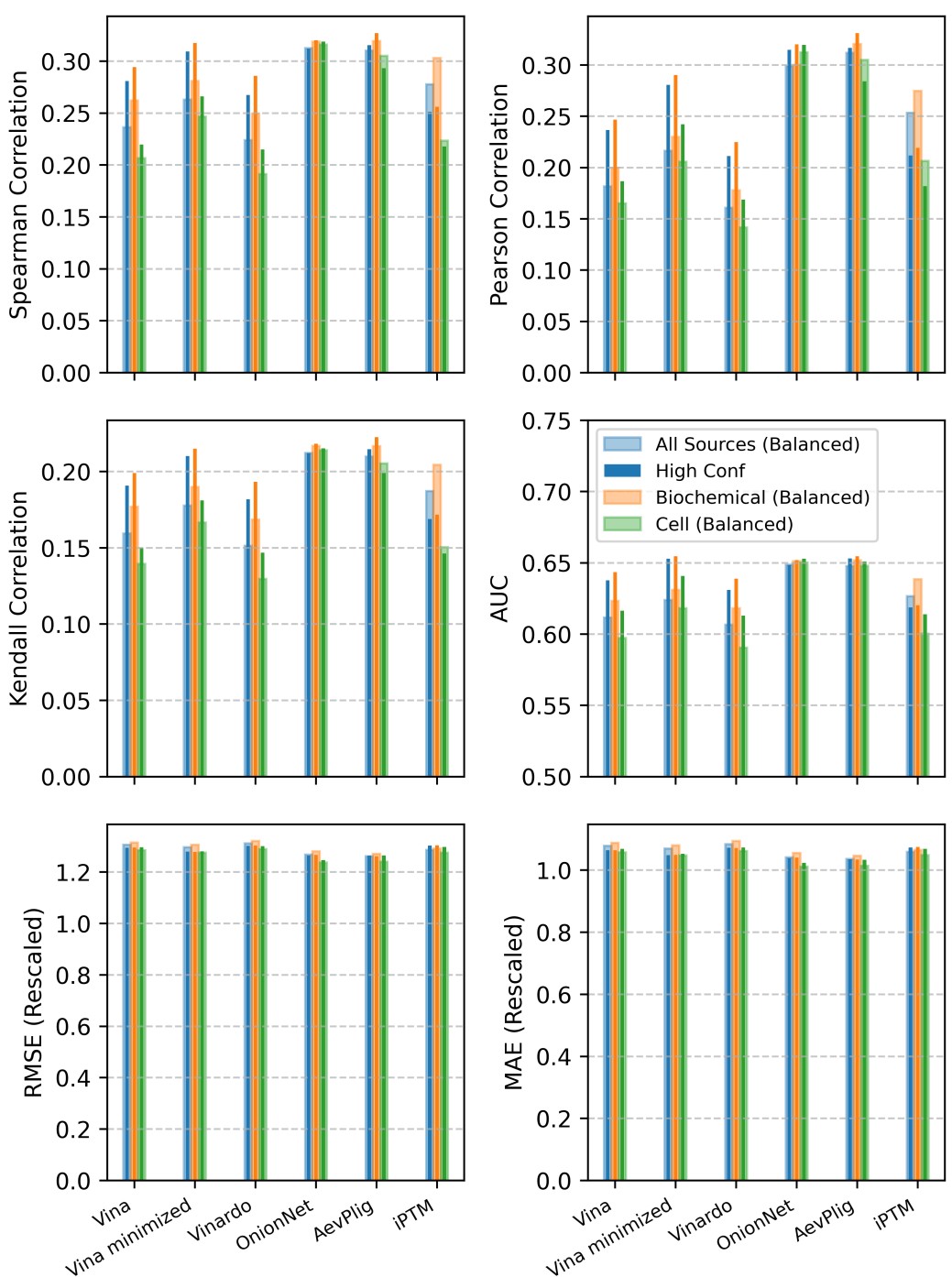

Figure 8: **Family-Balanced Performance Benchmarks.** Comparison of binding affinity prediction performance using weighted metrics. The "All Sources (Balanced)" and "Biochemical (Balanced)" bars report metrics calculated using sample weights inversely proportional to protein family frequency. The consistency with the main benchmarks suggests the model performance is robust to family distribution biases.

## H    USE OF LARGE LANGUAGE MODELS

Large Language Models were used to proofread and improve earlier versions of the manuscript.

