# OpenReview forum: "SAIR: Enabling Deep Learning for Protein-Ligand Interactions with a Synthetic Structural Dataset"
_ICLR.cc/2026/Conference — ICLR 2026 Poster_

### Official Review · Reviewer_947p · 2025-10-27

**Soundness:** 3
**Presentation:** 3
**Contribution:** 3
**Rating:** 6
**Confidence:** 4

**Summary:**

This paper introduces the Structurally Augmented IC50 Repository (SAIR), a dataset of over 1 million unique protein-ligand pairs with associated activity data curated from ChEMBL and BindingDB. The 3D structures for these complexes were computationally generated using the Boltz-1x co-folding model. The paper presents a characterization of the dataset, an evaluation of the structural quality using PoseBusters, and benchmarks for several binding affinity prediction models on the new data.

**Strengths:**

1. The dataset's size is a significant contribution that may contribute to addressing the data scarcity problem in structure-based modeling. The authors' commitment to making the dataset publicly available is highly commendable.
2. The use of a standard validation tool (PoseBusters) to confirm the physical plausibility of the computationally generated structures adds credibility and provides users with confidence in the data's quality.
3. The analysis correlating Boltz-1x confidence metrics (e.g., iPTM) with experimental binding affinity is a novel and interesting finding. This is timely, and it echos recent observations about the potential of AlphaFold-like models to rank binding affinities[1].

[1] Hong X, Gao B, Jia Y, et al. How Good is AlphaFold3 at Ranking Drug Binding Affinities?[J]. bioRxiv, 2025: 2025.05. 27.656341.

**Weaknesses:**

**Major Concern**:
The use of PoseBusters is a commendable step for ensuring the physical and chemical plausibility of the generated structures. However, this validation does not—and cannot—confirm the biological correctness of the binding pose. Boltz-1x performs prediction from sequence, which is analogous to a "blind docking" experiment. This creates a significant risk that the model places a ligand in a physically plausible but biologically incorrect pocket, which would fundamentally mislead any downstream model.

For proteins with structures in PDB, I recommend performing structural alignments between their generated structures and known holo-structures to quantify pocket overlap and correctness. For the proteins without known structures—where the predictions are inherently more speculative as they fall outside Boltz's structural training domain—this increased uncertainty should be explicitly discussed. I recommend comparing the predicted pocket against annotated functional domains from UniProt or the known pockets of homologous proteins. Furthermore, some datasets constructed with template-docking methods (e.g., BindingNet v2[1]), which assume similar ligands bind in the same pocket and can give mostly correct pockets, should be more adequately discussed.

**Minor Concern**:
The term "assay" appeared many times in this paper. However, in the bioactivity prediction field, an "assay" is often defined as a set of measurements for a single target conducted under a consistent experimental protocol (often corresponding to a unique ChEMBL assay ID). To avoid ambiguity, I recommend using the more precise term "assay type"(binding, functional, ADMET, ...), "assay level"(cell, target, ...), or "measurement"(IC50, Ki, Kd). However, I still strongly recommend providing the ChEMBL assay ID and organizing data in BindingDB by assays following BindingNet[1] and ActFound[2]. Data from different assays, even if they report the same "assay type," can vary dramatically due to differences in experimental protocols (e.g., cell-based vs. biochemical), conditions (pH, temperature, cofactors), and instrumentation. Lumping these data together without distinction introduces severe batch effects, and there is also evidence for the benefit of considering "assay" in modeling [3].

[1] Zhu H, Li X, Chen B, et al. Augmented BindingNet dataset for enhanced ligand binding pose predictions using deep learning[J]. npj Drug Discovery, 2025, 2(1): 1.
[2] Feng B, Liu Z, Huang N, et al. A bioactivity foundation model using pairwise meta-learning[J]. Nature Machine Intelligence, 2024, 6(8): 962-974.
[3] Feng B, Liu Z, Li H, et al. Hierarchical affinity landscape navigation through learning a shared pocket-ligand space[J]. Patterns, 2025, 6(10).

**Questions:**

No questions.

---

> ### Author Response · Authors · 2025-11-19
> **Response to Reviewer 947p**
>
> We thank the reviewer for their support of the dataset's scale and for the helpful suggestions.
>
> > Major Concern: ...validation does not—and cannot confirm the biological correctness of the binding pose. ...Boltz-1x performs prediction from sequence, which is analogous to a "blind docking" experiment.
>
> We agree. PoseBusters validates chemical plausibility, not biological truth. We have added a "Limitations" subsection explicitly stating that SAIR is a source of training data, not ground-truth binding sites. We recommend users leverage the provided confidence heads (e.g., interaction_ptm) which correlate with bioactivity as a filter for biological plausibility.
>
> > For proteins with structures in PDB, I recommend performing structural alignments…
>
> A core design principle of SAIR was to exclude any (UniProt, Ligand) pairs present in the PDB to ensure the dataset is "leak-proof" for downstream tasks. Therefore, we cannot perform these alignments for the entries in SAIR.
>
> > Minor Concern: The term "assay" appeared many times in this paper. ...I recommend using the more precise term "assay type"... I still strongly recommend providing the ChEMBL assay ID…
>
> We appreciate this distinction. We have revised the manuscript to use "Assay Type" (e.g., binding, functional) more consistently. We will also add the ChEMBL assay IDs to the `sair.parquet` file, allowing users to perform their own granular grouping.

---

> > ### Comment · Reviewer_947p · 2025-11-21
> >
> > Thank you for your response and your contribution of releasing assay IDs. I would like to maintain my borderline score because my major concern regarding the risk of biologically incorrect binding pocket remains unresolved.

---

### Official Review · Reviewer_rhF2 · 2025-10-29

**Soundness:** 3
**Presentation:** 3
**Contribution:** 3
**Rating:** 6
**Confidence:** 4

**Summary:**

The paper introduces **SAIR**, a million-scale synthetic structure–activity dataset ($\approx$ 1.05M protein–ligand systems) built by curating bioactivity from ChEMBL and BindingDB, then generating 3D complexes using Boltz-1x co-folding. The curation removes entries with missing identifiers, validity flags, inequalities, and measurements outside a defined dynamic range; ligands are standardized (desalting, neutral-pH protonation, RDKit canonical SMILES). To avoid leakage from experimental structures, (UniProt, CCD) pairs present in PDB are excluded. The authors assess structural plausibility with PoseBusters, report overall low failure rates (only $\approx$ 0.53% of complexes have all five poses failing), analyze confidence metrics (iPTM, interaction-PTM, complex iPLDDT) vs potency, and benchmark a small set of affinity predictors (Vina/Vinardo, OnionNet-2, AEV-PLIG).

**Strengths:**

- **New dataset for proper demand.** If fully released, SAIR could be a valuable pretraining/analysis data source for structure-based binding affinity prediction models. Since the major limitation of training PLI models is lack of paired data, this approach is suitable in terms of its purpose and time.
- **Scale and transparency.** Clear end-to-end pipeline from bioactivity ingestion to synthetic structure generation; explicit de-duplication and PDB-leakage control. As the authors stated, the computational cost for making this data is very expensive and thus this dataset is invaluable.
- **Interesting diagnostics.** Introduction/usage of interface-focused confidence metrics (interaction-PTM, iPTM) and assay-aware correlation analyses (biochemical > cell).

**Weaknesses:**

Despite the strengths of the proposed dataset, my major concern is rooted in the fact that creating a synthetic structure-affinity dataset should minimize uncertainty in both structure and affinity.
1. **More rigorous validation of predicted structure.** All complexes are treated as monomers with canonical UniProt sequences, despite many targets (e.g., GPCRs) being oligomeric or using non-canonical constructs. Recent studies show that co-folding models can predict plausible-looking yet physically inconsistent pockets/poses (including persistent ligand placement after pocket disruption), yielding false positives in downstream analyses. [1,2] Thus, I suggest that the authors include an additional assessment of pocket validity. For instance, they could examine whether the predicted pocket residues are consistent with experimentally validated binding sites observed in highly similar structures associated with proteins with similar sequence, especially for the proteins from rare protein family.
2. **Potential distillation bias.** Because Boltz-1x uses deterministic MSA subsampling and results improve most on a high-confidence subset, the dataset may over-represent 'easy' (well-constrained/high-MSA or training-set–similar) proteins and under-represent rare/low-MSA targets. I would welcome analyses quantifying how confidence filtering shifts family/MSA composition and whether conclusions hold under family-balanced or MSA-balanced evaluations. I think this may results in making false positive data for rare protein family.
3. **Confidence wording vs effect size.** In the page 7's first and second paragraphs, calling $r_{s}\approx 0.25$ "strong" overstates the effect; the paper's own narrative elsewhere acknowledges low absolute correlations comparable to interface metrics.

[1] Masters, Matthew R., Amr H. Mahmoud, and Markus A. Lill. "Investigating whether deep learning models for co-folding learn the physics of protein-ligand interactions." _Nature Communications_ 16.1 (2025): 8854.
[2] Škrinjar, Peter, et al. "Have protein-ligand co-folding methods moved beyond memorisation?." _BioRxiv_ (2025): 2025-02.

**Questions:**

1. **Affinity aggregation strategy over five structures.** For each protein–ligand pair, do you report model performance per-structure, or do you aggregate (mean/min/best) across the five predicted structures?
2. **Pocket diversity in main text.** I saw that there are no information about pocket diversity in the main text despite its detailed explanation is provided in appendix. In my opinion, high pocket diversity for several proteins may introduce unintended bias in data. I'd like to know about the authors opinion of this issue. If this is problematic, giving diversity of protein pocket and let the users handling about that issue themselves can be a solution.
3. **Cofactors and tiny ligands.** How are cofactors/ions handled (e.g., heavy-atom count near 1, as described in table 3)? Are such entries filtered or annotated, and how do they affect pocket identification and affinity correlations?
4. **About the degradation of performance of OnionNet-2.** In Fig. 6, OnionNet-2 attains only Pearson r $\approx$ 0.35 on SAIR, whereas its reported performance on the CASF-2016 scoring set is r $\approx$ 0.864. Given that OnionNet-2 relies primarily on protein–ligand contact patterns, this large gap suggests potential issues with pocket localization/contacts in the co-folded structures.

---

> ### Author Response · Authors · 2025-11-19
> **Response to Reviewer rhF2**
>
> We thank the reviewer for their detailed analysis and helpful comments.
>
> > ...co-folding models can predict plausible-looking yet physically inconsistent pockets/poses... I suggest that the authors include an additional assessment of pocket validity.
>
> We agree that physical plausibility (low energy) does not guarantee biological correctness. However, since we explicitly filtered out proteins present in the PDB to prevent data leakage, we do not have ground truth structures to perform the RMSD/alignment checks the reviewer suggests. We can rely on the folding model’s confidence metrics (iPTM, iPLDDT), which we show correlate with experimental affinity. We have updated the manuscript to explicitly caution users that high confidence is a proxy, not a guarantee, of the correct biological pocket.
>
> > Cofactors and tiny ligands. How are cofactors/ions handled...?
>
>
> This is an important point, and we thank the reviewer for bringing it to our attention. In this version of SAIR, we folded the monomeric protein chain and the ligand of interest. We did not explicitly model cofactors. We have added a comment to clarify this. In addition, to address this, we will add a drug_like filter column to the dataset metadata (how do we do this?), allowing users to easily exclude tiny ligands (e.g., ions, fragments) that might be artifacts or cofactor-dependent.
>
>
> > About the degradation of performance of OnionNet-2. In Fig. 6, OnionNet-2 attains only Pearson r 0.35 on SAIR, whereas its reported performance on the CASF-2016 scoring set is r 0.864.
>
> This drop highlights the domain shift between clean crystal structures (CASF) and noisy synthetic structures (SAIR). OnionNet-2 relies on precise contact patterns. Its poor zero-shot performance on SAIR confirms that models trained on crystal structures do not generalize well to the inference setting (synthetic structures). This validates the need for SAIR: retraining models on SAIR (as the GatorAffinity authors have done, see response to all reviewers) allows them to adapt to this distribution.

---

> > ### Comment · Reviewer_rhF2 · 2025-11-28
> >
> > I appreciate the authors’ efforts in addressing the weaknesses raised in the original review. However, I remain concerned that the core issue regarding potential distillation bias has not been sufficiently addressed. Simply improving the quality and coverage of in-distribution datasets may not necessarily lead to better generalization toward broader, more diverse protein–ligand interaction spaces. A deeper explanation or empirical evidence showing robustness beyond the observed training distribution would significantly strengthen the claims.
> >
> > Additionally, the previous suggestion to refine and tone down certain explanations, particularly where the manuscript may overstate novelty or impact, still appears unresolved.
> >
> > Given these remaining concerns, I believe it is appropriate to reconsider the current evaluation score unless the authors can more clearly address these critical points in the revision.

---

> > > ### Author Response · Authors · 2025-11-28
> > > **Response to Reviewer rhF2**
> > >
> > > We thank the reviewer for their continued engagement. We take the concern regarding distillation bias very seriously. To address your specific request, we have performed two new quantitative analyses:
> > >
> > > 1.We analyzed how confidence filtering affects the distribution of protein families. As hypothesized by the reviewer, filtering for high-confidence structures does introduce a shift. The proportion of Kinases and Enzymes increases in the high-confidence subset (e.g., Kinases rise from $\approx 29\%$ to $\approx 35\%$). Conversely, GPCRs and Nuclear Receptors see a reduction in relative proportion, though they are still well-represented (thousands of structures). We have updated Figure 3 in the manuscript to explicitly overlay the "High Confidence" distribution against the "All Sources" distribution, making this bias transparent to the user, and added some text to present this findings.
> > >
> > > 2. To verify that our benchmark results are not artificially inflated by this over-representation of "easy" families (like Kinases), we re-ran our binding affinity benchmarks using a family-balanced evaluation. We re-computed the metrics, weighting samples inversely to the frequency of their protein family. This ensures that a rare target contributes as much to the final score as a common one. As shown in the new appendix G and Figure 8, the performance trends remain consistent under the balanced evaluation. While there is a minor variance in absolute numbers, the AEV-PLIG and OnionNet-2 models still outperform classical scoring functions, and the correlation between confidence metrics (iPTM) and affinity remains stable. This provides empirical evidence that the signal in SAIR is robust and not merely a result of overfitting to over-represented, well-constrained protein families.
> > >
> > > In regards to toning down claims, we apologize for missing this comment in the previous version. We have carefully reviewed the manuscript to tone down subjective language as requested: We removed the word "strong" when describing correlations of $r_s \approx 0.25$. We now describe these as "positive correlations" that provide a "predictive signal".

---

### Official Review · Reviewer_gzGf · 2025-11-01

**Soundness:** 3
**Presentation:** 2
**Contribution:** 3
**Rating:** 6
**Confidence:** 3

**Summary:**

This paper introduces SAIR, the largest dataset of 3D protein-ligand structures with annotated binding affinity data. The dataset comprises over 1 million unique protein-ligand systems and a total of 5.2 million structures, was created by computationally folding complexes from the ChEMBL and BindingDB databases using the Boltz-1x model. The primary goal of SAIR is to provide a comprehensive resource for training and evaluating structure-based deep learning models in drug discovery.

**Strengths:**

1. The paper addresses an important real-world problem in structure-based drug discovery, i.e., lacking of large-scale protein-ligand 3D structural data with bioactivity annotations.
2. The authors provide detailed descriptions of the specific steps for filtering and processing data from ChEMBL and BindingDB.
3. The authors provide comprehensive and insightful experimental analysis.

**Weaknesses:**

1. One core conclusion of this work is that models trained on experimental structures show modest correlations on SAIR's synthetic structures, which the authors attribute to data distribution differences. However, it would be better to include an important validation experiment: fine-tuning or training (from scratch) a model on a SAIR subset, then evaluating on a held-out SAIR test set. Demonstrating significant performance improvement would more convincingly prove SAIR's value as a "training resource." Currently, the paper positions SAIR more as a "benchmark exposing existing model limitations" rather than demonstrating its potential for "enabling new model development".

2. The generated dataset relies on the structural quality generated by Boltz-1x. While the authors validated using PoseBusters with a low failure rate (~3%), complete dependence on a single generative model may introduce biases specific to that model. For instance, the model might underperform on specific protein families. It would be beneficial to explore these risks more thoroughly in the discussion.

3. Fig. 4 shows that "internal energy" is the primary cause of PoseBusters validation failures. Does this suggest the generated conformations may not be physically reasonable?

**Questions:**

See Weaknesses

---

> ### Author Response · Authors · 2025-11-19
> **Response to Reviewer gzGf**
>
> We thank the reviewer for their helpful comments and constructive feedback.
>
> > ...it would be better to include an important validation experiment: fine-tuning or training (from scratch) a model on a SAIR subset, then evaluating on a held-out SAIR test set. Demonstrating significant performance improvement would more convincingly prove SAIR's value as a "training resource."
>
> As pointed out in the response to all reviewers, the recent GatorAffinity paper demonstrates that pre-training on SAIR improves RMSE on the PDBbind benchmark from 1.343 (Kd+Ki only) to 1.293 (Kd+Ki+SAIR), confirming SAIR’s value as a training resource. We have added a section on this to the main text. In addition, we would be happy to perform further experiments if the reviewer thinks it would improve the submission.
>
> > ...complete dependence on a single generative model may introduce biases specific to that model. For instance, the model might underperform on specific protein families.
>
> This is a valid point. SAIR is a distilled dataset and carries the inductive biases of Boltz-1x. We have added a "Limitations" section explicitly stating this. To help users identify potential collapse or bias, we have added the "pocket diversity" statistics to the dataset metadata (Appendix E), allowing users to filter out proteins where the model fails to generate diverse binding modes. We would like to thank the reviewer for this valuable suggestion. If the reviewers request, we could add a limited assessment of the boltz-1x model’s performance on pocket prediction. For instance, for the samples classified as a kinase with ligands known to be kinase inhibitors, this could be a check that the predicted pocket is the kinase pocket.
>
> > Fig. 4 shows that "internal energy" is the primary cause of PoseBusters validation failures. Does this suggest the generated conformations may not be physically reasonable?
>
>
> First of all, we would like to highlight that only around 3% of the generated structures fail the posebusters test. Figure 4. Numbers are out of those 3% (we have added a comment to clarify that). That aside, high internal energy often indicates minor steric clashes or bond length anomalies common in raw generative outputs, rather than fundamental structural failures. These can often be resolved via standard energy minimization (e.g., with OpenMM) without altering the binding pose significantly. We have clarified this distinction in Section 3.2.

---

### Official Review · Reviewer_VX3P · 2025-11-01

**Soundness:** 3
**Presentation:** 3
**Contribution:** 3
**Rating:** 4
**Confidence:** 4

**Summary:**

The manuscript introduces SAIR, a large-scale dataset of protein–ligand 3D complexes paired with activity labels. Activity data are curated from ChEMBL and BindingDB, and corresponding complex structures are computationally generated using the Boltz-1x model—this synthetic-structure generation is the core contribution. The work contributes: (1) a large, publicly available synthetic 3D structure–activity resource built with Boltz-1x from ChEMBL/BindingDB, (2) quality analysis highlighting failure modes, and (3) an initial benchmarking suite motivating methods calibrated to this data regime.

As an initial benchmark, they compare empirical scoring functions (Vina, Vinardo), a 3D CNN (OnionNet-2), and a GNN (AEV-PLIG). While ML models outperform classical scoring, correlations with ground-truth affinities remain modest, underscoring the need for models fine-tuned to synthetic structure distributions.

**Strengths:**

1.  Constructs the largest dataset of predicted protein–ligand 3D complexes paired with activity labels; if released, it would be a valuable community resource for training and benchmarking.

2.  Provides a preliminary benchmarking of multiple models on the dataset, offering a baseline reference for future methods.

**Weaknesses:**

1) Model choice: The use of Boltz-1 for complex generation is not justified against more robust contemporaries (e.g., AlphaFold 3).

2) Benchmarking impact: Current results do not show that the dataset materially improves performance of existing models.

3) Baselines and finetuning: Baselines omit modern 3D algorithms; no fine-tuning of foundation models (e.g., Boltz-2) or comparisons of frozen vs. full training.

4) 3D utility ablations: No comparisons to sequence-/2D-only models or to randomized/alternative poses to assess sensitivity to 3D quality and pose source.

5) IC50 Values from diverse assay conditions (system, temperature, readout) are pooled without harmonization or consistency analysis.

6) Over-reliance on correlation/AUC; missing RMSE/MAE and decision-centric metrics (e.g., enrichment, hit rate).

7) Experimental clarity: Dataset splitting (protein/scaffold/time) and leakage prevention strategies are not specified.

**Questions:**

1.  Why was Boltz-1 selected over more robust alternatives (e.g., AlphaFold 3)

2.  Show evidence that training on SAIR improves performance vs. existing datasets (e.g., pretrain/fine-tune on SAIR vs. not)


3.  Will the authors include stronger modern 3D baselines and fine-tune foundation models (e.g., Boltz-2), and compare frozen vs. full fine-tuning?

4.  Add ablations comparing sequence-only, 2D-only, randomized poses, and alternative pose generators to quantify sensitivity to 3D quality and pose source.


5.  Will the authors report RMSE/MAE, enrichment/hit-rate metrics, and calibration with confidence intervals?

6.  The exact split protocols (by protein/scaffold/time), identity/scaffold thresholds, preprocessing steps, and measures to prevent leakage

---

> ### Author Response · Authors · 2025-11-19
> **Response to Reviewer VX3P**
>
> We thank the reviewer for their assessment and very helpful comments.
>
> > Model choice: The use of Boltz-1 for complex generation is not justified against more robust contemporaries (e.g., AlphaFold 3).
>
> As discussed in the response to all reviewers, we chose the best model for protein-ligand systems, that was available with an open-source license at the time of doing this work.
>
> > Benchmarking impact: Current results do not show that the dataset materially improves performance of existing models.
>
> As also mentioned in the response to all reviewers, the updated version of the manuscript points to the recent "GatorAffinity" preprint , which independently validates that pre-training on SAIR leads to SOTA performance on PDBbind.
>
> > Baselines omit modern 3D algorithms; no fine-tuning of foundation models (e.g., Boltz-2) or comparisons of frozen vs. full training.
>
> For the stronger modern 3D baselines, it is our understanding that AEV-PLIG and OnionNet are some of the best models for 3D protein-ligand affinity prediction. However, we would be happy to add more, if the reviewer has some specific models in mind.
>
> In terms of fine-tuning foundation models, this is something we are working on, but it is unfortunately unlikely that we will be able to do it before the discussion time ends
>
>
> > IC50 Values from diverse assay conditions (system, temperature, readout) are pooled without harmonization or consistency analysis.
>
> We appreciate this point. We do hope that our filtering strategy does somehow tackle this. In addition, while assay noise exists, large-scale pre-training often benefits from data quantity even with noise, as observed in the GatorAffinity study where adding the noisier IC50 data (SAIR) to Kd/Ki data improved performance. We have added some text to acknowledge this in the revised version of the paper. We are also adding Chembl IDs to the dataset metadata, so users can do their own filtering.
>
> > Over-reliance on correlation/AUC; missing RMSE/MAE and decision-centric metrics (e.g., enrichment, hit rate).
>
> Updated Benchmarks (Figure 6): We have updated Figure 6 to include panels for RMSE and MAE.
> Methodology: To ensure a scientifically valid comparison between Machine Learning models (which predict pIC50) and empirical scoring functions (which output energies in kcal/mol or confidence probabilities), we applied a linear recalibration to all model outputs before calculating error metrics. This maps the scores to the pIC50 scale of the ground truth, allowing for a direct "apples-to-apples" comparison of error rates.
> Results: The new panels confirm that the ML baselines (AevPlig, OnionNet-2) achieve lower absolute errors compared to classical scoring functions (Vina, Vinardo), even after the latter are optimally scaled. This reinforces the conclusion that training on bioactivity data provides a distinct advantage over purely physics-based scoring for this task.
> Since our benchmark dataset consists of experimental affinity data rather than a constructed "active vs. decoy" screening library, calculating explicit Enrichment Factors (EF) is not standard. However, the AUC metric serves as the direct mathematical proxy for this decision-making capability

---

### Author Response · Authors · 2025-11-19
**Response to all Reviewers**

We thank all reviewers for their constructive feedback and for recognizing the potential of SAIR as a large-scale resource. We have uploaded a revised manuscript addressing the suggestions regarding clarity, terminology, and baselines. Below, we address the four primary questions raised across reviews:

**1. Training Models on SAIR** A primary question was whether SAIR materially improves model performance when used for training. During the review period, a preprint titled "GatorAffinity: Boosting Protein-Ligand Binding Affinity Prediction with Large-Scale Synthetic Structural Data" was released by an independent research group. This work explicitly utilizes the SAIR dataset for pre-training. The study reports that pre-training on SAIR combined with BindingDB data allows the model to outperform current SOTA methods (including GIGN and PSICHIC) on the PDBbind benchmark. Furthermore, the study explicitly demonstrates a "data scaling law," showing that increasing the size of the synthetic pre-training data (provided by SAIR) consistently lowers the Root Mean Square Error (RMSE) on the test set. This serves as independent, objective validation that SAIR is a highly effective resource for training high-performance affinity models. We have added a small section discussing this to the paper. However, if the reviewers request it, we are opening to retraining AEV-PLIG, OnionNet, or another model on our data

**2. Choice of Boltz-1x over AlphaFold 3** Another recurrent question was why Boltz-1x was selected over AlphaFold 3 (AF3). A core goal of SAIR is to be a community resource. At the time of generation, the full training code and weights for AF3 were not available under a license that permitted the generation and redistribution of a dataset of this scale (5M+ structures). Boltz-1x is open-source (MIT license), ensuring the dataset and the pipeline remain reproducible. In addition, Boltz-1x is competitive with AF3 in metrics such as LDDT, DOCKQ and RMSD, but has much better physical validity (see figures 5 and 6 of https://www.biorxiv.org/content/10.1101/2024.11.19.624167v3.full.pdf), making it arguably the best existing model for protein-ligand systems. We did not use Boltz-2, as it was not available at the time, but there is no significant difference between Boltz-1x and Boltz-2 for the protein-ligand modality. We have added these arguments to the manuscript, and we thank the reviewers for bringing this to our attention.

**3. Robustness and Distillation Bias**: To address concerns that the dataset might be biased toward "easy" targets (e.g., Kinases) over-represented in the training set, we performed two new analyses: First, we explicitly analyzed the distribution shift caused by confidence filtering (new Figure 3), transparently showing the enrichment of Kinases/Enzymes in the high-confidence subset.Family-Balanced Benchmarks: Second, we re-ran our binding affinity benchmarks using a family-balanced evaluation (new Appendix G), where samples are weighted inversely to their protein family frequency. The performance trends remain consistent (e.g., AEV-PLIG Spearman correlation remains stable at $\approx 0.31$), providing empirical evidence that the predictive signal in SAIR is robust and generalizes across diverse protein space, rather than being driven solely by over-represented families.

**4.Updated Benchmarks & Metrics**: RMSE & MAE: We have updated Figure 6 to include RMSE and MAE metrics. To ensure a scientifically valid comparison between ML models (predicting pIC50) and empirical scoring functions (predicting energy/confidence), we applied a linear recalibration to all outputs before calculating error metrics. The ML baselines (OnionNet-2, AEV-PLIG) achieve lower absolute errors compared to classical scoring functions, reinforcing the value of training on bioactivity data.

---

### Meta-Review · Area_Chair_GyhQ · 2026-01-06

**Summary:**

1. *947p* major concern was that PoseBusters cannot be used to validate the biological correctness of the binding pose.  *rhF2* has a similar concern.
2. *rhF2* also thinks the data may oversample easy high-MSA proteins.
3. *gzGf* and *VX3P* would like to see if models trained or fine-tuned on SAIR's synthetic structures improve in performance.
4.  *gzGf* and *VX3P* questioned reliance on the Boltz-1x generative model.
5. *VX3P* noted lack of 3D algorithms and fine-tuned foundation models
6. *VX3P* suggests ablating 3D features
7. *VX3P* notes the use of diverse assay conditions in the dataset
8. *VX3P* wanted more metrics

**Reviewer Concerns:**

1. The authors explicitly note this limitation.  The reject the reviewers suggestion to use PBD to keep the dataset pure for downstream tasks.
2. The authors filtered for confidence to confirm the bias and reran evaluation after balancing.
3. The authors point to a recent preprint showing SAIR training can boost performance.
4. The authors acknowledge this limitation and note they chose Boltz-1x since it was open license.
5. The authors believe they have included modern 3D algorithms, but note fine-tuning of foundations models was not possible in time.
6. not addressed
7. The authors acknowledge the point and indicate diverse conditions can improve performance in downstream tasks.
8. The authors added metrics

**Reviewer Scores:**

- *947p* explicitly noted they would keep their score of 6
- *rhF2* explicitly noted that they needed the bias concern to be addressed. There is a reasonable chance the authors action addressed their concern and they would raise their score 6 to 8.
- *gzGf* there is a small chance they would increase their score 6 to 8.
- *VX3P* there is a small chance they would increase their score 4 to 6.

---

### Decision · Program_Chairs · 2026-01-26

Accept (Poster)